# Optimization of the Electro-Peroxone Process for Micropollutant Abatement Using Chemical Kinetic Approaches

**DOI:** 10.3390/molecules24142638

**Published:** 2019-07-20

**Authors:** Huijiao Wang, Lu Su, Shuai Zhu, Wei Zhu, Xia Han, Yi Cheng, Gang Yu, Yujue Wang

**Affiliations:** 1Department of Chemical Engineering, Tsinghua University, Beijing 100084, China; 2School of Environment, Beijing Key Laboratory for Emerging Organic Contaminants Control, State Key Joint Laboratory of Environmental Simulation and Pollution Control, Tsinghua University, Beijing 100084, China; 3Hangzhou Jinhong Real Estate Co., Ltd., Hangzhou 310000, China; 4Beijing Guohuan Tsinghua Environmental Engineering Design & Research Institute Co., Ltd., Beijing 100084, China; 5Sinopec Energy and Environmental Engineering Co., Ltd., Wuhan 430070, China

**Keywords:** electrocatalytic ozonation, electro-peroxone, model, ozone, pharmaceutical, water treatment

## Abstract

The electro-peroxone (E-peroxone) process is an emerging electrocatalytic ozonation process that is enabled by in situ producing hydrogen peroxide (H_2_O_2_) from cathodic oxygen reduction during ozonation. The in situ-generated H_2_O_2_ can then promote ozone (O_3_) transformation to hydroxyl radicals (•OH), and thus enhance the abatement of ozone-refractory pollutants compared to conventional ozonation. In this study, a chemical kinetic model was employed to simulate micropollutant abatement during the E-peroxone treatment of various water matrices (surface water, secondary wastewater effluent, and groundwater). Results show that by following the O_3_ and •OH exposures during the E-peroxone process, the abatement kinetics of a variety of model micropollutants could be well predicted using the model. In addition, the effect of specific ozone doses on micropollutant abatement efficiencies could be quantitatively evaluated using the model. Therefore, the chemical kinetic model can be used to reveal important information for the design and optimization of the treatment time and ozone doses of the E-peroxone process for cost-effective micropollutant abatement in water and wastewater treatment.

## 1. Introduction

The electro-peroxone (E-peroxone) process is an emerging electrocatalytic ozonation process that has shown great potential for micropollutant abatement in water and wastewater treatment [1,2,3,4]. Unlike most catalytic ozonation processes that rely on the direct reaction of ozone (O_3_) with catalysts to produce hydroxyl radicals (•OH) [5,6,7,8,9,10], the E-peroxone process employs an indirect catalytic strategy to enhance •OH production during ozonation—that is, by electrocatalytically producing hydrogen peroxide (H_2_O_2_) from cathodic oxygen (O_2_) reduction (Equation (1)) during ozonation. The in situ generated H_2_O_2_ can then react with O_3_ to form •OH via the so-called peroxone reaction (Equation (2)) [11]. Due to the enhanced O_3_ transformation to •OH by electro-generated H_2_O_2_, the E-peroxone process can considerably accelerate organic pollutant degradation, especially ozone-refractory pollutants (e.g., ibuprofen, 1,4-dioxane, and oxalic acid) compared to conventional ozonation [1,12,13,14,15]. In addition, the electro-generated H_2_O_2_ can quickly quench hypobromous acid (a key intermediate for bromate formation in ozone-based processes) back to bromide [16,17,18]. Therefore, the E-peroxone process can effectively inhibit bromate formation during the treatment of bromide-containing water, which has been a major concern associated with conventional ozonation [19,20,21,22]. Thus, by using O_2_ that is always present in excess in ozone-based processes to produce H_2_O_2_, the E-peroxone process can significantly improve the performance of water and wastewater treatment in many aspects [23,24].
(1)$$O_{2}+2H^{+}+2e^{-}\to H_{2}O_{2}$$
(2)$$2H_{2}O_{2}+2O_{3}\to H_{2}O+3O_{2}+{\mathrm{HO}}_{2}•+•\mathrm{OH}$$


Design and optimization of the E-peroxone process for micropollutant abatement in real water matrices are challenging [23]. Real water matrices (e.g., natural water and wastewater) usually contain a large number of structurally diverse micropollutants [25,26,27]. The abatement kinetics of the various micropollutants differ dramatically during ozone-based processes (including the E-peroxone process). For example, micropollutants with high ozone reactivity can typically be quickly eliminated in several seconds during ozone-based processes, while micropollutants with low ozone reactivity usually require significantly longer treatment time (several minutes to tens of minutes) [3,25,28,29,30]. Moreover, ozone doses required to abate the various micropollutants also differ significantly. For the same abatement objective (e.g., 90% abatement), significantly lower specific ozone doses are required for micropollutants that have high ozone reactivity than those that have low ozone reactivity (e.g., <0.5 mg O_3_/mg dissolved organic carbon (DOC) vs. >1.0 mg O_3_/mg DOC) [28,31,32,33]. To ensure the effective abatement of various micropollutants, while minimizing the time and financial cost of water treatment, operational parameters of the E-peroxone process (e.g., ozone doses, currents, and treatment time) need to be carefully optimized [3,20].

Knowledge of the abatement kinetics of various micropollutants is critical for designing and optimizing the E-peroxone process for water treatment [23]. Due to the large numbers of micropollutants present in real water matrices, it is impractical to experimentally measure the abatement kinetics of each micropollutant during water treatment [31,33]. In contrast, chemical kinetic models may provide a feasible way to predict the abatement kinetics of various micropollutants during the E-peroxone process [28,33,34,35,36,37]. In a previous study [32], we have found that the final abatement efficiency of various micropollutants at the end of the E-peroxone process can be reasonably predicted using a simple chemical kinetic model (Equation (3)):
(3)$$\frac{C}{C_{0}}=e^{-\left(k_{O_{3}}\int [O_{3}]\mathrm{dt}+k_{•\mathrm{OH}}\int [•\mathrm{OH}]\mathrm{dt}\right)}$$
where C_0_ and C are the concentration of a micropollutant at treatment time 0 and t; *k*_O3_ and *k*_•OH_ are the second-order rate constants for the reaction of micropollutants with O_3_ and •OH, respectively; and $\int \left[O_{3}\right]\mathrm{dt}$ and ∫[•OH]dt are the overall O_3_ and •OH exposures observed at the end of the E-peroxone treatment.

Therefore, it is expected that by monitoring the time-wise variation of O_3_ and •OH exposures during the E-peroxone process, the model may allow a generalized prediction of the abatement kinetics of various micropollutants. This can provide necessary kinetic information to optimize the treatment time of the E-peroxone process for micropollutant abatement. In addition, the model may offer a useful tool to quantitatively assess the relationship between ozone doses, the oxidation capacity of O_3_ and •OH, and micropollutant abatement efficiency during the E-peroxone process, thus providing valuable information for optimizing ozone doses for cost-effective micropollutant abatement [32,34,35,37,38].

The main objective of this study was to evaluate the feasibility of the chemical kinetic model for kinetic modelling of micropollutant abatement during the E-peroxone process. Several micropollutants (i.e., diclofenac (DA), gemfibrozil (GF), bezafibrate (BF), ibuprofen (IBU), clofibric acid (CA), and para-chlorobenzoic acid (*p*-CBA)) that have varying O_3_ reactivities (*k*_O3_ ranging from ≤0.15 to 6.8 × 10^5^ M^−1^ s^−1^) were selected as model compounds and spiked in three real water samples (surface water, secondary wastewater effluent, and groundwater). The water samples were then treated by the E-peroxone process. The abatement kinetics of the various micropollutants were simulated using the model based on O_3_ and •OH exposures observed during the course of the E-peroxone treatment. The effect of ozone doses on micropollutant abatement was evaluated using the chemical kinetic model. The results suggest that the model may provide a valuable tool for the design and optimization of the E-peroxone process for micropollutant abatement.

## 2. Results and Discussion

### 2.1. Kinetic Modelling of Micropollutant Abatement by the E-Peroxone Process

Micropollutant abatement in the selected surface water by conventional ozonation (current = 0 mA) and the E-peroxone process with varying applied currents (10–50 mA) are shown in Figure 1. Based on the time-wise O_3_ and •OH exposures observed during the treatment processes (Figure 2a,b, see Appendix A for more information), the abatement kinetics of micropollutants are simulated using the chemical kinetic model (Equation (3)) and shown as dash lines in Figure 1. As shown, micropollutants with varying ozone reactivities were abated at significantly different rates during the treatments, and the chemical kinetic model satisfactorily simulated the abatement kinetics of the various micropollutants (R^2^ = 0.92 for the linear correlation of experimentally measured and model predicted abatement efficiency, see Appendix A for more information).

During the E-peroxone process, micropollutants can be abated by chemical oxidation with O_3_ and/or •OH in the bulk solution, as well as electrochemical oxidation at the anode surface [13,23,24]. Due to the low concentrations of micropollutants, the rate of micropollutant abatement by electrochemical oxidation is usually limited by their mass transfer to the anode surface [3,39]. As shown in Figure 2c, the concentrations of the various micropollutants were abated by only ~0.5–2% during 2 min of electrolysis treatment of the surface water. In contrast, they were generally completely abated in the same time period during the E-peroxone process (Figure 1). These comparisons indicate that the contribution of electrochemical oxidation for micropollutant abatement can be neglected during the E-peroxone process, and micropollutants are abated by essentially chemical oxidation with O_3_ and/or •OH (see Appendix A for more detailed discussion) [3,32]. Therefore, the abatement kinetics of micropollutants during the E-peroxone process can be reasonably simulated using the model (Equation (3)) based on only O_3_ and •OH exposure observed during the treatment.

Figure 2a,b show that increasing applied currents from 0 to 50 mA resulted in slower increases of O_3_ exposure, but faster increases of •OH exposure during the E-peroxone process. These changes can be attributed to the acceleration of O_3_ transformation to •OH by electro-generated H_2_O_2_ [12,32]. As shown in Appendix A, increasing applied currents increased almost linearly the rate of H_2_O_2_ electro-generation at the cathode, which in turn increased the rate of peroxone reaction of O_3_ with H_2_O_2_ to form •OH. Due to the faster kinetics of O_3_ transformation to •OH at higher currents, •OH exposures increased more rapidly while O_3_ exposures increased more slowly as the applied currents were progressively increased during the E-peroxone process. These changes caused different effects on the abatement kinetics of micropollutants that have varying ozone reactivities (Figure 1).

As shown in Figure 1a,b, DA and GF were quickly eliminated within 10 s (the first sampling point) during both conventional ozonation and the E-peroxone process. Due to their high ozone rate constants (*k*_O3_ = 6.8 × 10^5^ and 5 × 10^4^ M^−1^ s^−1^), only very small O_3_ exposures (6.8 × 10^−6^ and 9.2 × 10^−5^ M s) were needed to abate the two compounds to below their detection limits (~99% abatement) according to the chemical kinetic model. These small O_3_ exposure requirements can be easily achieved within a few seconds of all treatment processes (see Figure 2a). Therefore, although there were significant decreases in the O_3_ exposures observed for the E-peroxone process compared to conventional ozonation, DA and GF were similarly eliminated within 10 s during all treatment processes (Figure 1a,b).

Compared with DA and GF, the abatement of BF, IBU, CA, and *p*-CBA required longer treatment time, especially during conventional ozonation (Figure 1c–f). Unlike the ozone-reactive DA and GF, BF, IBU, CA, and *p*-CBA have relatively low ozone rate constants (*k*_O3_ ranging from ≤0.15 to 590 M^−1^ s^−1^), and they were abated predominantly by •OH oxidation during ozonation and the E-peroxone treatment (see Appendix A). Corresponding to the faster increase of •OH exposure at higher currents, the abatement kinetics of BF, IBU, CA, and *p*-CBA increased with increasing applied currents. Therefore, they were eliminated in significantly shorter treatment time during the E-peroxone process (1–2 min) than during conventional ozonation (e.g., 20 min for CA and *p*-CBA, data not shown).

The abatement kinetics of micropollutants during the E-peroxone process (30 mA) with varying specific ozone doses (0.5–1.5 mg O_3_/mg DOC) are shown in Figure 3. DA and GF were similarly eliminated within 10 s for all three tested specific ozone doses (0.5, 1.0, and 1.5 mg O_3_/mg DOC, Figure 3a,b), which can be attributed to the small requirement of O_3_ exposure for the elimination of ozone-reactive micropollutants [28,31,32]. In contrast, 0.5 mg O_3_/mg DOC was insufficient for the elimination of BF, IBU, CA, and *p*-CBA, which were abated only by ~38–67% during the E-peroxone process (Figure 3c–f). The abatement efficiencies of BF, IBU, CA, and *p*-CBA then increased significantly to generally ≥90% with increasing the specific ozone dose to 1.0 mg O_3_/mg DOC. Therefore, further increase of the specific ozone dose to 1.5 mg O_3_/mg DOC may not be cost-effective for micropollutant abatement in the selected surface water, taking into account the marginal increase of micropollutant abatement efficiency and linear increase of energy demand for ozone generation (~15 kWh/kg O_3_ [25,40]).

Similar to what has been shown in Figure 1, the abatement kinetics of micropollutants during the E-peroxone treatment with varying specific ozone doses can be well simulated using the chemical kinetic model (Figure 3 and Appendix A). These observations indicate that by monitoring the time-wise variations of O_3_ and •OH exposures, a generalized prediction of the abatement kinetics of various micropollutants during the E-peroxone process is possible using the chemical kinetic model. This kinetic information can then be used to optimize the treatment time (or hydraulic residence time) of the E-peroxone process for micropollutant abatement.

### 2.2. Modelling the Effect of Ozone Doses on Micropollutant Abatement

The results shown in Figure 3 suggest that there is a marginal benefit associated with the application of higher ozone doses in the E-peroxone process. To get more insight into this effect, the relationships between ozone doses, O_3_ and •OH oxidation capacity, as well as micropollutant abatement efficiency during the E-peroxone treatment of various water matrices (surface water, secondary wastewater effluent, and groundwater) are evaluated using the chemical kinetic model. 

Figure 4a,b show that for all three selected water matrices, the overall O_3_ and •OH exposures observed at the end of E-peroxone process increased linearly with increasing specific ozone doses in the range of 0.5–1.5 mg O_3_/mg DOC. Similar results have been reported in previous ozonation treatment of municipal wastewater effluents [33]. These observations suggest that for a specific water matrix, increasing ozone doses in the typical range applied in water and wastewater treatment can lead to almost linear increases of the oxidation capacity of O_3_ and •OH during the E-peroxone process. This, in turn, resulted in linear increases in the overall $k_{O_{3}}\int [O_{3}]{\mathrm{dt}+k}_{•\mathrm{OH}}\int [•\mathrm{OH}]\mathrm{dt}$ value (referred to as O_ke_ value hereafter) of the various micropollutants observed at the end of the E-peroxone treatment of the three water matrices (see Figure 5 insets for GF and CA, see Appendix A insets for the other micropollutants).

However, as the model shows, the abatement efficiency of micropollutant (C/C_0_) is an exponential function of the O_ke_ value (see the simulation curve in Figure 5). Therefore, the increase of specific ozone doses will not linearly increase the abatement efficiency of micropollutants during the E-peroxone process. Specifically, the model predicts that the value of C/C_0_ will decrease rapidly from ~0.9 to ~0.1 in the O_ke_ value range of ~0.1–2.3, whereas the value of C/C_0_ will not change considerably outside this narrow range of O_ke_ value. For ozone-reactive DA and GF, their O_ke_ values are considerably higher than 2.3 during the E-peroxone treatment of the three water matrices with even the lowest specific ozone dose (0.5 mg O_3_/mg DOC, see Figure 5a–c and Appendix A). Therefore, the model predicts that DA and GF can be similarly completely abated in all three water matrices by the E-peroxone process at the varying specific ozone doses of 0.5–1.5 mg O_3_/mg DOC. This model prediction agrees well with the experimental observations (see Figure 5a–c and Appendix A).

In contrast to DA and GF, the increase of specific ozone doses resulted in different effects on the abatement of micropollutants that have low ozone reactivity in the three water matrices. For the selected surface water, increasing specific ozone doses from 0.5 to 1.0 mg O_3_/mg DOC increased the O_ke_ value of CA from 0.51 to 2.3, where C/C_0_ of 0.6 and 0.1 will be obtained according to the model. The simulation result suggests that 1.0 mg O_3_/mg DOC is sufficient for the effective abatement of CA in the selected surface water, whereas further increasing the specific ozone dose can only marginally increase the abatement efficiency of CA. Consistent with the model prediction, Figure 5d shows that the experimentally measured C/C_0_ decreased considerably from 0.54 to 0.07 as specific ozone doses were increased from 0.5 to 1.0 mg O_3_/mg DOC during the E-peroxone treatment of the selected surface water. However, further increasing the specific ozone dose to 1.5 mg O_3_/mg DOC only slightly decreased the C/C_0_ value of CA to 0.02 due to the non-linear relationship between the O_ke_ value and abatement efficiency of micropollutants. For the selected secondary wastewater effluent and groundwater, the increase of specific ozone doses from 0.5 to 1.5 mg O_3_/mg DOC increased the O_ke_ value of CA from ~0.30 to 1.85 and from 0.21 to 1.00, respectively. According to the model, these increases in the O_ke_ value will lead to a significant decrease in C/C_0_ from 0.74 to 0.16 and from 0.81 to 0.37, respectively, which is in agreement with the experimental observations (Figure 5e,f).

Overall, the results shown in Figure 5 suggest that the effect of ozone dose on micropollutant abatement efficiency is highly dependent on the ozone reactivity of micropollutant, as well as the water matrix. Based on the linear regression equation between specific ozone dose and the O_ke_ value observed for the various micropollutants during the E-peroxone treatment of the three water matrices (insets of Figure 5 and Appendix A), the specific ozone doses required to achieve 90% abatement (i.e., C/C_0_ = 0.1) of the various micropollutants tested in this study are calculated using the chemical kinetic model (Equation (3)) and shown in Figure 6.

Interestingly, the specific ozone doses required to abate 90% of ozone-reactive DA and GF differ only slightly for the three water matrices (~0.14–0.32 mg O_3_/mg DOC). In contrast, the specific ozone doses required to abate 90% of ozone-refractory micropollutants (e.g., IBU and CA) are significantly different for the three water matrices. These observations can possibly be attributed to the different abatement mechanisms of ozone-reactive and ozone-refractory micropollutants during the E-peroxone process. Because of their high ozone reactivity, ozone-reactive micropollutants are abated by primarily O_3_ oxidation during the E-peroxone process (see Appendix A) [25,32,41]. Water matrices influence the abatement of ozone-reactive micropollutants mainly through the competing consumption of O_3_ with background water constituents, especially with ozone-reactive DOM [25,32]. Normalizing applied ozone doses to the DOC value of the water matrix will mask the influence of the water matrix to some extent, thus resulting in similar specific ozone dose requirement for ozone-reactive micropollutant abatement in different water matrices [25].

Unlike ozone-reactive micropollutants, ozone-refractory micropollutants are abated predominantly by •OH oxidation during the E-peroxone process (see Appendix A). It is well-known that background water constituents (e.g., DOM and carbonate) have significant effects on both •OH generation from O_3_ decomposition and •OH scavenging [25]. Due to their different water constituents, the •OH yield (moles of •OH produced per mole of O_3_) varied considerably during the E-peroxone treatment of the selected surface water, secondary effluent, and groundwater (~26%, ~17%, and ~31%, respectively) [32]. In addition, the •OH scavenging rate also differed significantly for the three water samples (1.3 × 10^5^, 2.6 × 10^5^, and 1.1 × 10^5^ s^−1^ for the selected surface water, secondary effluent, and groundwater) [32]. Therefore, significantly different specific O_3_ doses are required to abate ozone-refractory micropollutants during the E-peroxone treatment of the three water matrices. 

Overall, Figure 6 shows that for the same removal objective, significantly higher ozone doses (and thus energy demand) are required to abate ozone-refractory micropollutants than ozone-reactive ones during the E-peroxone process. Therefore, it is more costly to remove ozone-refractory micropollutants than ozone-reactive ones in water and wastewater treatment. Figure 7 shows the model-predicted specific ozone doses that are required to abate *p*-CBA (the most ozone-resistant micropollutant tested in this study) to varying degrees during the E-peroxone treatment of the three water matrices. As shown, required specific ozone doses increase gradually as abatement efficiencies increase from 0 to 80%. However, further increase the abatement efficiency, especially beyond 90%, will require a sharp increase in specific ozone doses, which will result in significant increases in the energy demand for ozone generation (~15 kWh/kg O_3_). This result suggests that water utilities have to carefully evaluate the trade-off between micropollutant abatement efficiencies and water treatment costs when applying the E-peroxone process, and the model may provide a useful tool for this purpose.

### 2.3. Implications

The results presented above indicate that micropollutants with differing ozone reactivity are abated at significantly different rates during the E-peroxone process. Moreover, ozone doses required to abate the various micropollutants are highly dependent on the ozone reactivity of micropollutants, as well as the water matrix. Therefore, the design and optimization of the E-peroxone process for micropollutant abatement in real water matrices is a challenging task. During the E-peroxone process, ozone-reactive micropollutants can generally be rapidly eliminated in various water matrices in much shorter treatment time (e.g., <10 s) than typical hydraulic residence time applied in water and wastewater treatment (~10–30 min). Therefore, the abatement kinetics of ozone-reactive micropollutants usually do not need to be taken into account when optimizing the E-peroxone process for micropollutant abatement. In contrast, the treatment time and ozone doses required to abate ozone-resistant micropollutants differ significantly depending on many process and water parameters. As a result, the abatement kinetics of ozone-resistant micropollutants have to be evaluated on a case-by-case basis for process optimization. Based on the O_3_ and •OH exposures observed during the E-peroxone process, the chemical kinetic model allows a generalized prediction of the abatement kinetics of various micropollutants. The ozone doses required to abate the various micropollutants to a certain degree can also be estimated using the model. Therefore, the chemical kinetic model may provide a useful tool for water utilities to design and optimize operational parameters (e.g., treatment time and ozone dose) of the E-peroxone process for effective and efficient micropollutant abatement in water and wastewater treatment.

## 3. Materials and Methods

### 3.1. Chemicals and Water Samples

DA, GF, BF, IBU, CA, and *p*-CBA with purity >98% were purchased from Sigma-Aldrich (Louis, MO, USA, see Appendix A for their *k*_O3_ and *k*_•OH_) [25,35,42,43]. *p*-CBA was used as the •OH probe for characterizing •OH exposure during the E-peroxone process because of its low reaction rate with O_3_ (k_O3_
*≤* 0.15 M^−1^ s^−1^) [35]. Potassium indigo trisulfonate (80–85%) and phosphoric acid (85%) were also purchased from Sigma-Aldrich. All other chemicals (e.g., sodium thiosulfate) were of analytical grade and purchased from Beijing Chemical Works Co., Beijing, China. All solutions were prepared with Milli-Q ultrapure water (resistivity > 18.2 MΩ cm).

Three real water matrices were used for ozonation and E-peroxone treatment in this study: A groundwater, a surface water collected from a reservoir in Beijing, and a secondary wastewater effluent from a municipal wastewater treatment plant in Beijing. After collection, the water samples were immediately stored in a refrigerator (4 °C), and then used within two weeks for experiments. The main water quality parameters are summarized in Appendix A. The three waters were used directly for E-peroxone treatment without addition of any electrolytes. Since the background concentrations of the tested micropollutants were below the detection limit of the analytical method (Appendix A), micropollutant stocks were spiked into the surface water to achieve an initial concentration of ~150 μg/L for each compound, which is close to the highest concentration of micropollutants detected in most real water matrices (natural water and municipal wastewater).

### 3.2. E-Peroxone Treatment of the Water Samples Containing Micropollutants

Batch tests of the E-peroxone treatment of water samples were conducted in a sealed glass column reactor (250 mL) at 15 ± 1 °C. Concentrated ozone stock solutions were prepared by continuously bubbling ozone-containing oxygen through Milli-Q water cooled in a water bath (15 °C). Small volumes of freshly prepared ozone stock solution were spiked into the water sample to achieve a specific O_3_ dose of 0.5–1.5 mg O_3_/mg DOC for the E-peroxone experiment. Simultaneously with spiking the O_3_ stock solutions, a DC power supply was turned on to supply constant currents (10, 30, and 50 mA) to the electrodes. The anode was a platinum plate (2 cm × 2 cm), and the cathode was a carbon-polytetrafluoroethylene (carbon-PTFE) electrode prepared with Vulcan XC-72 carbon powder (Cabot Corp., Boston, MA, USA), PTFE dispersion, and anhydrous alcohol (2 cm × 5 cm) [44]. The treatments were stopped when the added ozone was completely depleted during the E-peroxone process. All presented results were based on duplicate experiments, with data variability less than 8%.

### 3.3. Analytical Methods

Samples were collected at preset time intervals during the treatments. H_2_O_2_ concentrations were analyzed using the potassium titanium (IV) oxalate method [45]. Aqueous O_3_ concentrations were analyzed with the indigo method [46]. O_3_ exposure (∫[O_3_]dt) was obtained by integrating the O_3_ decay over reaction time [47] and used to represent the oxidation capacity of O_3_ during the treatment [25,32]. •OH exposure (∫[•OH]dt) was back calculated from the abatement of the •OH probe *p*-CBA [33] and used to represent the oxidation capacity of •OH during the treatment [25,32]. For the analysis of *p*-CBA and other micropollutants, the collected samples were immediately quenched with sodium thiosulfate, followed by 0.22 μm membrane filtration (Jinteng, Tianjin, China). Then water samples were analyzed with an ultra-performance liquid chromatography/tandem mass spectrometry (API 3200 LC/MS/MS system, Applied Biosystems, Foster, CA, USA). The separation of target compounds was performed with a Waters XBridgeTM C18 column (3.0 × 150 mm, 3.5 μm, Milford, MA, USA). The mobile phase consisted of methanol (solvent A) and 10 mM ammonium acetate in ultrapure water (solvent B) at a flow rate of 0.35 L/min. The mass spectrometer was operated in negative electrospray ionization (ESI) and multiple reaction monitoring mode (see Appendix A for more details).

## 4. Conclusions

The results of this study demonstrate that the chemical kinetic model can provide a useful tool to design and optimize the E-peroxone process for cost-effective and energy-efficient micropollutant abatement in water and wastewater treatment. By monitoring the variations of O_3_ and •OH exposures during the E-peroxone process, the abatement kinetics of various micropollutants can be well predicted using the model. This can provide important information to optimize the treatment time (or hydraulic residence time) of the E-peroxone process for micropollutant abatement. In addition, the specific ozone doses required to abate micropollutants to a certain degree (e.g., 90% abatement) can be estimated using the model. This may offer a valuable tool for water utilities to balance between micropollutant abatement objectives and operation costs of water and wastewater treatment, thus optimizing ozone doses of the E-peroxone process for real applications.

## Figures and Tables

**Figure 1 molecules-24-02638-f001:**
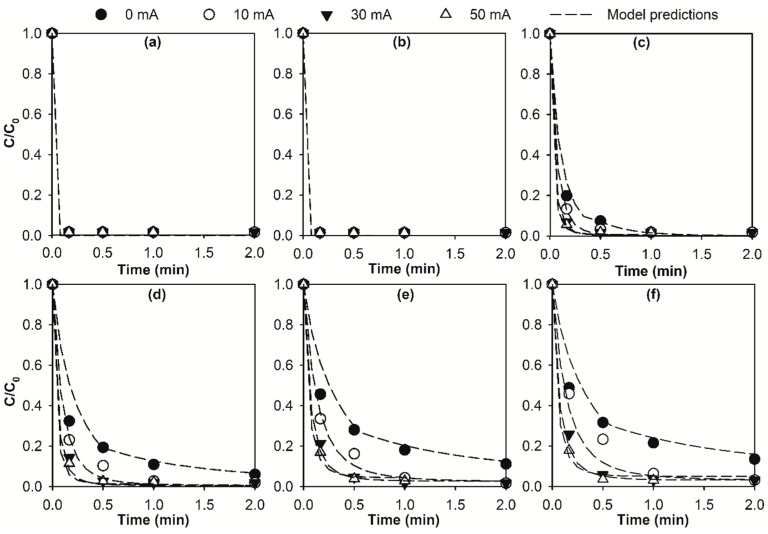
Abatement of (**a**) diclofenac (DA), (**b**) gemfibrozil (GF), (**c**) bezafibrate (BF), (**d**) ibuprofen (IBU), (**e**) clofibric acid (CA), and (**f**) para-chlorobenzoic acid (*p*-CBA) during the electro-peroxone (E-peroxone) treatment of surface water at different currents (0 mA (ozonation), 10 mA, 30 mA, and 50 mA). The symbols in the plot represent experimental data, and short dash lines are model predictions. (Reaction conditions: Each micropollutant concentration ~150 μg/L, and specific ozone dose = 1.5 mg O_3_/mg dissolved organic carbon (DOC)).

**Figure 2 molecules-24-02638-f002:**
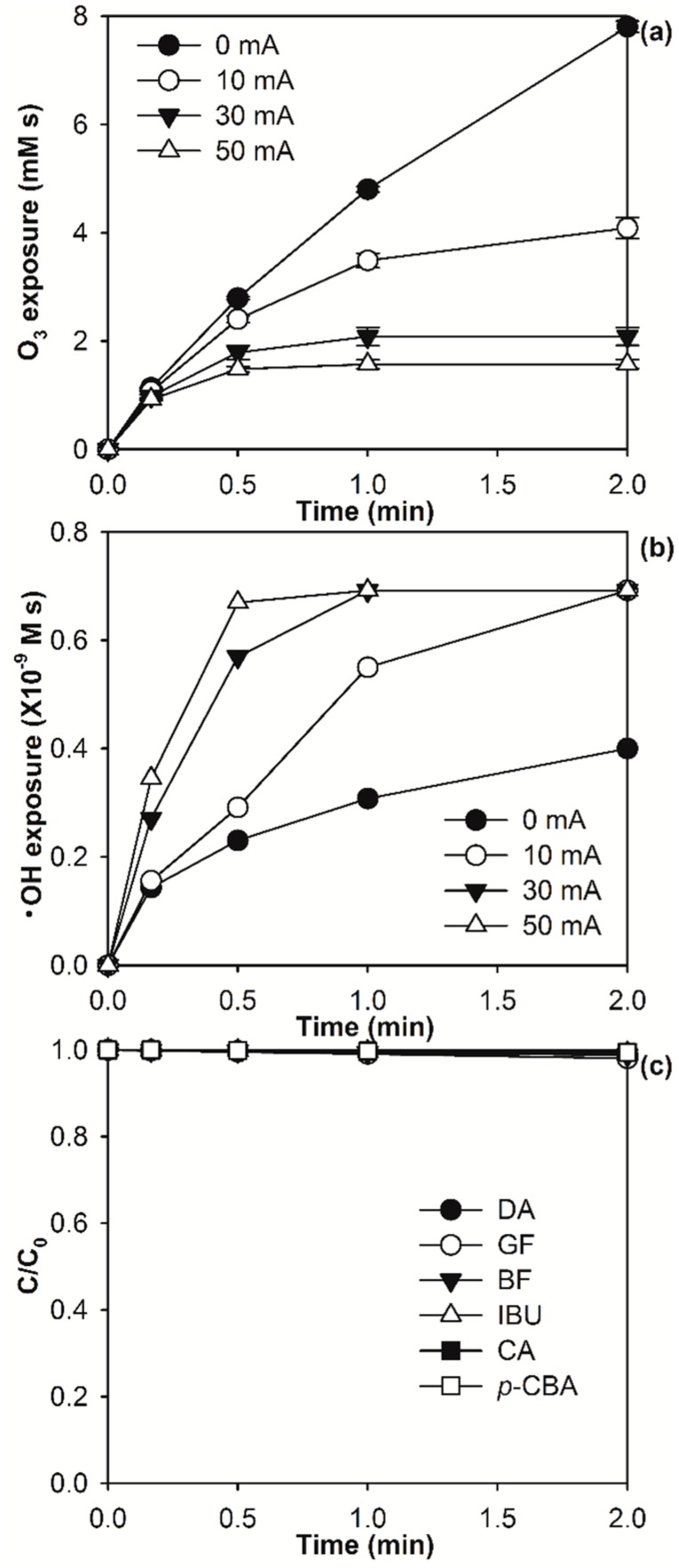
(**a**) Ozone (O_3_) and (**b**) hydroxyl radicals (•OH) exposures during the E-peroxone treatment of surface water at different currents (0 mA (ozonation), 10 mA, 30 mA, and 50 mA), and (**c**) micropollutant abatements from surface water by electrolysis at the current of 30 mA. (Reaction conditions: Each micropollutant concentration ~150 μg/L, and specific ozone dose = 1.5 mg O_3_/mg dissolved organic carbon (DOC)).

**Figure 3 molecules-24-02638-f003:**
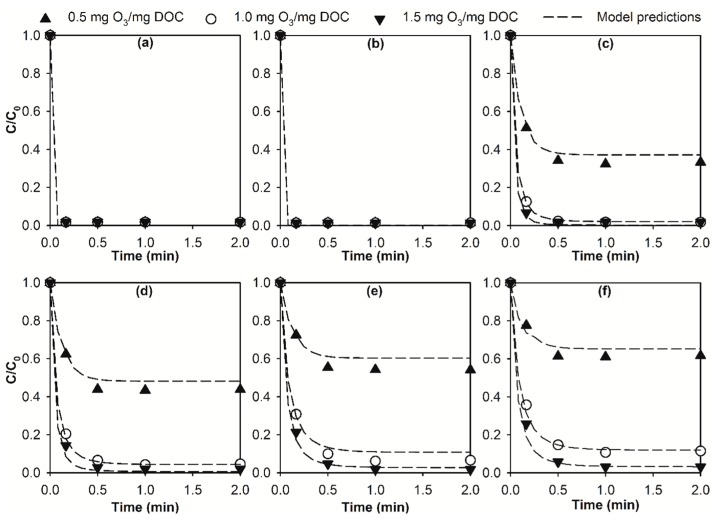
Abatement of (**a**) diclofenac (DA), (**b**) gemfibrozil (GF), (**c**) bezafibrate (BF), (**d**) ibuprofen (IBU), (**e**) clofibric acid (CA), and (**f**) para-chlorobenzoic acid (*p*-CBA) during the E-peroxone treatment of surface water at different specific ozone dose (0.5, 1.0, and 1.5 mg O_3_/mg dissolved organic carbon (DOC)). The symbols in the plot represent experimental data, and short dash lines stand for model predictions. (Reaction conditions: Each micropollutant concentration ~150 μg/L, and current = 30 mA).

**Figure 4 molecules-24-02638-f004:**
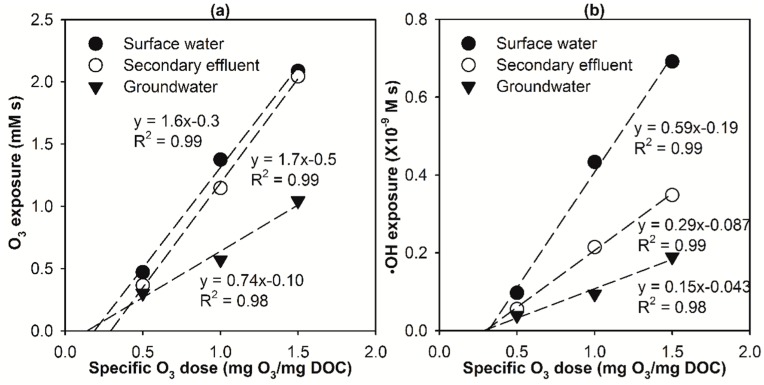
Linear regression of (**a**) O_3_ and (**b**) •OH exposure as a function of specific ozone (O_3_) dose during E-peroxone treatment of surface water, secondary effluent, and groundwater. (Reaction conditions: Each micropollutant concentration ~150 μg/L, and current = 30 mA).

**Figure 5 molecules-24-02638-f005:**
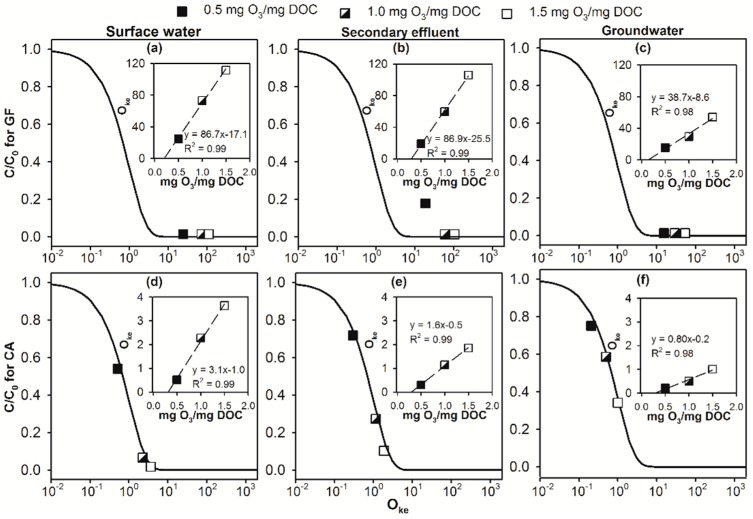
Abatement efficiency of (**a**–**c**) gemfibrozil (GF) and (**d**–**f**) clofibric acid (CA) as a function of their O_ke_ values observed during the E-peroxone treatment of (**a**,**d**) surface water, (**b**,**e**) secondary effluent, and (**c**,**f**) groundwater with varying specific ozone (O_3_) doses. The symbols in the plots represent experimentally measured results; the solid lines are model simulation using Equation (3). The inset plot shows linear regression between specific O_3_ dose and the O_ke_ value observed for selected micropollutant during the E-peroxone process. (Reaction conditions: Current = 30 mA, and each micropollutant concentration ~150 μg/L).

**Figure 6 molecules-24-02638-f006:**
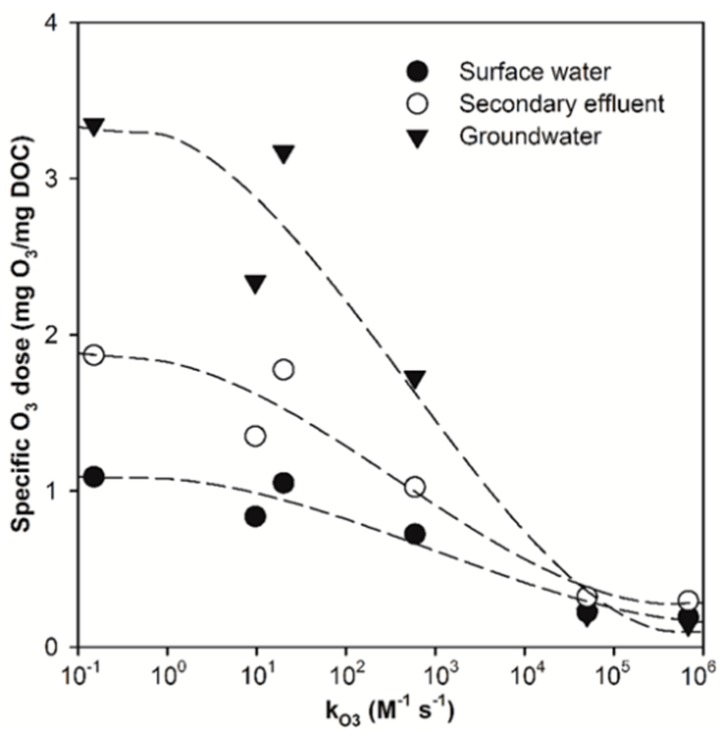
Model-predicted specific ozone (O_3_) dose required to achieve 90% abatement efficiency of the selected micropollutants during the E-peroxone treatment of surface water, secondary effluent, and groundwater.

**Figure 7 molecules-24-02638-f007:**
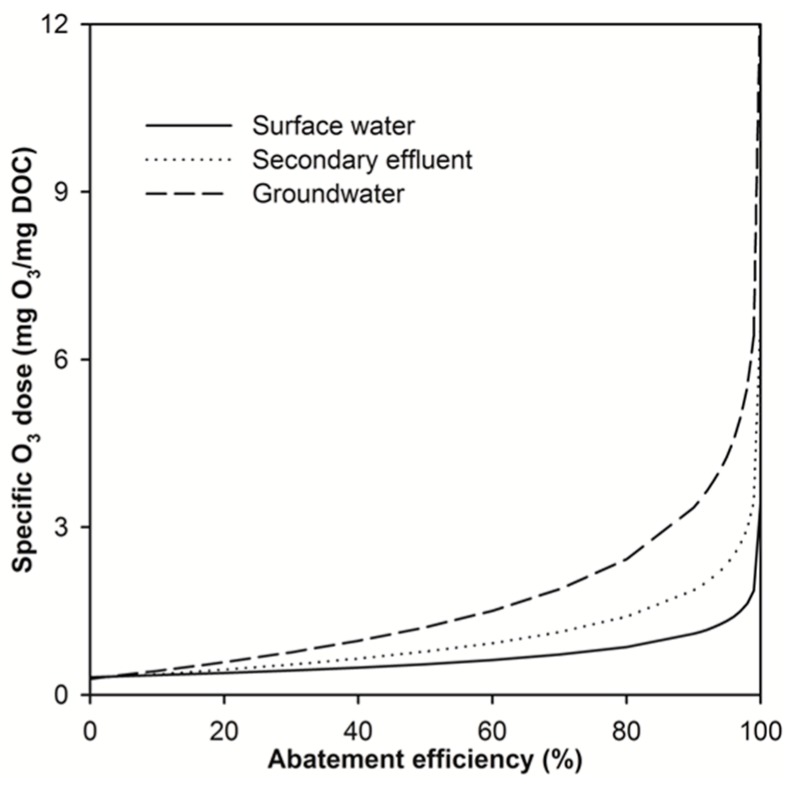
The required specific ozone (O_3_) dose as a function of *p*-CBA abatement efficiency during the E-peroxone treatment of different water matrices. Results were obtained based on the linear regression equation of specific O_3_ doses with O_ke_ values in Appendix A inset plots, as well as the chemical kinetic model.

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
