# Peer review of "Optimization of the Electro-Peroxone Process for Micropollutant Abatement Using Chemical Kinetic Approaches"

_molecules, 2019, doi:10.3390/molecules24142638_

Round 1

Reviewer 1 Report

The manuscript is very well written and results are clearly presented. The experimental part is very well carried out. As a minor point I would like authors to perform some degradation test of the compounds which are rapidly (less than 10 seconds) by the e-peroxone process with a higher initial drug concentration in order to check the quality of the model with more experimental points. 

After this minor revision the paper can be published.

Reviewer 2 Report

Dear Editor,

Regarding the manuscript "Optimization of the electro-peroxone process for micropollutant abatement using chemical kinetic approaches" by Wang et al. I think it is a good paper and sounds really god for your readers.

Material and Methods generally appears before results. I saw you put M&M after results. Is it a journal guideline?

My suggestion is about statistics: Can you use R2 in nonlinear regressions? Even if possible, Adjusted R2 would be better, as your models have different variables.More variables are added, greater will be the R2. So, Adjusted R2 correct this issue.

Sincerely yours. 
